# Efficient and Approximate Per-Example Gradient Norms for Gradient Noise Scale

**Gavia Gray**
Cerebras Systems
Toronto, Canada
gngdb.labs@gmail.com

**Anshul Samar**
Cerebras Systems
Sunnyvale, CA
anshul@cerebras.net

**Joel Hestness**
Cerebras Systems
Sunnyvale, CA
joel@cerebras.net

## Abstract

Gradient Noise Scale (GNS) is valuable to compute because it provides a suggestion for a compute efficient batch size during training: small enough to be compute efficient and large enough to take advantage of parallelism. While it can be a valuable tool, computing GNS is often cumbersome or expensive due to the difficulty of obtaining gradient norms over a small batch of examples (smaller than the training batch used). An existing trick for collecting "efficient" per-example gradient norms is inefficient in transformer or convolutional models. By assuming activations are normally distributed, we compute an approximate per-example gradient norm that tracks the true per-example gradient norm in practical settings. Using this approximation, we construct a Scaled Output Gradient Noise Scale (SOGNS) that is generally applicable at negligible cost and provides additional feedback to the practitioner during training.

## 1 Introduction

Gradient Noise Scale (GNS) correlates with the *critical batch size*, which is the point below which one may expect to linearly accelerate training by adding examples to the batch (McCandlish et al., 2018). For this reason, the batch size prescribed by GNS has been demonstrated to be useful while training GPT-3 (Brown et al., 2020).

Computing the GNS requires gradient norms from small and large batches (described in Section 2). However, in settings where we desire high performance compute, batch sizes typically need to be large, making it difficult or costly to sample small batch gradients. Goodfellow (2015) introduces a trick to access per-example gradient norms efficiently, but this trick cannot be applied in settings with tensor rank larger than 2. In particular, transformer language models have rank-3 tensor with batch, sequence and hidden dimensions. To address this problem, we construct an approximation that assumes normally distributed activations at layer inputs, which allows us to access per-example norms efficiently (described in Section 3.1), and provide a reference implementation: `https://github.com/CerebrasResearch/nanoGNS`.

## 2 Background

McCandlish et al. (2018) suggest using the "simple" GNS, $\mathcal{B}_{\text{simple}}$[1], as a metric to inform the practitioner while training a model,

$$\mathcal{B}_{\text{simple}} = \frac{tr(\Sigma)}{G^T G}$$

---

[1]This approximation is denoted as "simple" because it assumes that the Hessian is diagonal in the Taylor expansion of the loss.

Workshop on Advancing Neural Network Training at 37th Conference on Neural Information Processing Systems (WANT@NeurIPS 2023).

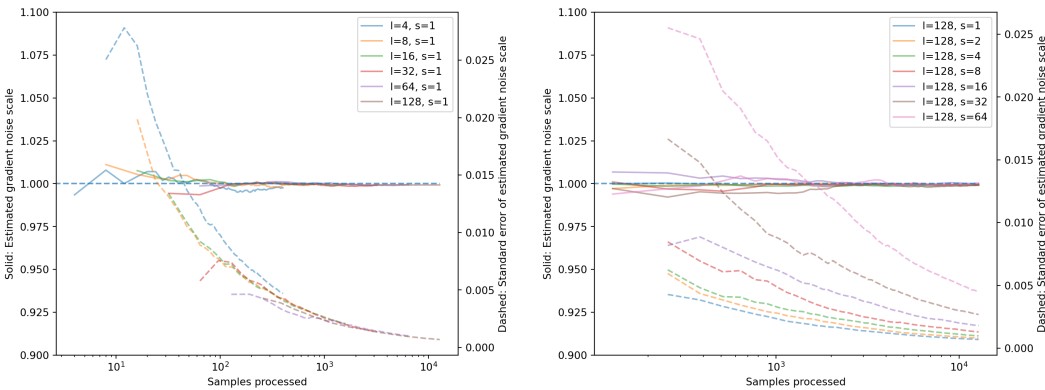

Figure 1: The variance of the GNS estimator for different $B_{\text{big}}$ (left) and $B_{\text{small}}$ (right) sizes. $B_{\text{big}} = l$ and $B_{\text{small}} = s$ in legends.

where $G$ are the gradients and $\Sigma$ is their associated covariance matrix. To compute $\mathcal{B}_{\text{simple}}$ McCandlish et al. (2018) further define the unbiased estimators $\mathcal{S}$ and $|\mathcal{G}|^2$ shown in Equations 1 and 2, where $B_{\text{big}}$ and $B_{\text{small}}$ are the batch sizes used to compute the gradients.

$$|\mathcal{G}|^2 := \frac{1}{B_{\text{big}} - B_{\text{small}}} \left( B_{\text{big}} |G_{B_{\text{big}}}|^2 - B_{\text{small}} |G_{B_{\text{small}}}|^2 \right) \approx G^T G \tag{1}$$

$$\mathcal{S} := \frac{1}{1/B_{\text{small}} - 1/B_{\text{big}}} \left( |G_{B_{\text{small}}}|^2 - |G_{B_{\text{big}}}|^2 \right) \approx tr(\Sigma). \tag{2}$$

We can easily compute $|G_{B_{\text{big}}}|$ using the accumulated gradients immediately after the backward pass. However, the challenge in computing $|G_{B_{\text{small}}}|$ is that it requires the gradients for a batch size that is smaller than the batch size used for the optimizer step. McCandlish et al. (2018) propose using the gradients communicated between Distributed Data Parallel (DDP) nodes but this means that the variance of the resulting GNS estimate is tied to that DDP configuration. A taxonomy of the options for computing $|G_{B_{\text{small}}}|$ is presented in Appendix A.

As $|G_{B_{\text{small}}}|$ may be estimated as the mean over samples within the minibatch, in accordance with the law of large numbers, the variance of the estimate decreases with the number of observations of the gradient norm. As shown in Figure 1, this implies the small batch size should be as small as possible to obtain an estimate of $|G_{B_{\text{small}}}|$, and thus the GNS, with minimal variance. Further discussion of this result may be found in Appendix B and code in Appendix B.1.

## 3 Efficient Per-example Gradient Norms

Goodfellow (2015) proposes a trick to compute gradient norms for individual examples in a minibatch, which would provide the minimum variance estimate of the GNS as described in Section 2. He observes that the squared norm of the gradient is a sum of elements in an outer product that can be factored into two smaller sums on the input vectors, eliminating the need to calculate the full outer product. It may be stated as follows using Einstein and Lagrange notation[2],

$$n_b^2 = (w')_{bik}^2 = x_{bi} x_{bi} y'_{bk} y'_{bk},$$

where $x$ are the activations prior to a linear layer, $y'$ are the gradients of the loss with respect to the outputs of the linear layer and $w'$ are the gradients of the loss with respect to the weights of the linear layer.

For networks of only linear layers acting on 2D inputs, this trick is sufficient to provide accurate GNS estimates[3]. However, for networks with convolutional or 3D inputs to linear layers, such as

---

[2]Further explanation of this notation may be found in Appendix C.

[3]As far as we are aware, this observation has not been made in prior work.

transformers, this trick is no longer efficient. For three dimensions, inputs $\mathbf{X} \in \mathbb{R}^{B \times T \times I}$ and outputs $\mathbf{Y} \in \mathbb{R}^{B \times T \times K}$ (Li et al., 2022), the per-example gradient norm $n_b$ is,

$$n_b^2 = (w')_{bik}^2 = \left( \sum_t x_{bti} y'_{btk} \right)^2 = x_{bti} y'_{btk} x_{bui} y'_{buk},$$

which has $O(T^2)$ complexity in the sequence length $T$. In these cases, computing the $w'$ explicitly, as the per-example gradient trick avoids, is more efficient. More details on this case are provided in Appendix C.1.

## 3.1 Proposed Approximation

Assuming all entries of $\mathbf{X}$ are IID Gaussian with a batch-dependent standard deviation $\sigma_b$ and mean zero allows us to compute the following expectation in closed form:

$$E\left[\sum_i x_{bi} x_{bi}\right] = \sum_i E[x_{bi} x_{bi}] = \sum_i \sigma^2(x_{bi}) = I \sigma^2(x_{bi}).$$

Appying this in the 3D case,

$$E[n_b^2] = E\left[ y'_{btk} y'_{buk} x_{bti} x_{bui} \right] = y'_{btk} y'_{buk} E\left[ x_{bti} x_{bui} \right] = \sum_{t,k} y'_{btk} y'_{buk} \sum_i \sigma_b^2 = I\sigma_b^2 \sum_{t,k} y'_{btk} y'_{buk}$$

and we know $\sigma_b^2 = \frac{1}{TI} \sum_{t,i} x_{bti} x_{bti}$ in line with our assumptions above, assuming $x_{bti}$ is zero-mean.

Factorizing the quadratic in the $t, u$ dimension produces

$$E[n_b^2] = I\sigma_b^2 \sum_k \left( \sum_t y'_{btk} \right)^2.$$

In practice, this says we can approximate $n_b$ as follows to construct $\eta_b$, the approximate per-example gradient norm,

$$n_b^2 \approx \eta_b^2 = I\sigma_b^2 \sum_k \left( \sum_t y'_{btk} \right)^2 = \left( \frac{1}{T} \sum_{t,i} x_{bti} x_{bti} \right) \sum_k \left( \sum_t y'_{btk} \right)^2,$$

and we can see that this is equal to the exact per-example gradient when $T = 1$:

$$\eta_b^2 = I\sigma_b^2 \sum_k \left( \sum_t y'_{btk} \right)^2 = I \frac{1}{I} \sum_i x_{bi} x_{bi} \sum_k (y'_{bk})^2 = x_{bi} x_{bi} y'_{bk} y'_{bk} = n_b^2$$

Experiments in Section 4, along with simulations in Appendix D, confirm that this approximation is accurate. This approximation may also be extended to $|G_{B_{\text{big}}}|$ as described in Appendix E, but this observation is unnecessary for the results presented here, as we assume the exact $|G_{B_{\text{big}}}|$ is easy to access.

Substituting $\eta_b^2$ into Equations 1 and 2 yields $\mathcal{B}_{\text{SOsimple}}$, the Scaled Output Gradient Noise Scale (SOGNS). The analogous metric using the exact per-example norm is $\mathcal{B}_{\text{PEPsimple}}$ the Per-Example Parameter Gradient Noise Scale (PEPGNS) (see Appendix C.2 for index of terms).

## 4 Experiments

### 4.1 Approximate Per-Example Gradient Noise Scale

We investigate how well SOGNS correlates with the observed GNS by training a 1M parameter Convolutional Neural Network (CNN) on MNIST (code linked in Appendix F). Figure 2a shows the overall fit of SOGNS to PEPGNS at all points throughout training for only the convolutional layers (the remaining linear layers only process 2D tensors so the estimate is exact). Throughout training, the relationship between SOGNS and PEPGNS is extremely regular at different orders of magnitude.

We also demonstrate the overall performance of the approximation by comparing the relationship between observed GNS and training loss. In Figure 2b, we replicate McCandlish et al. (2018) and plot $\mathcal{B}_{\text{crit}}$ as the authors measured. We see that the correlation to the critical batch size is similar for both SOGNS and PEPGNS. Additional statistics are plotted in Appendix F.

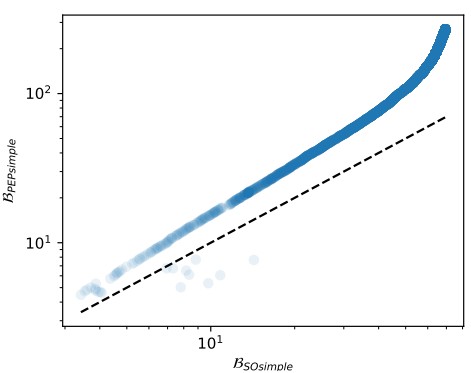

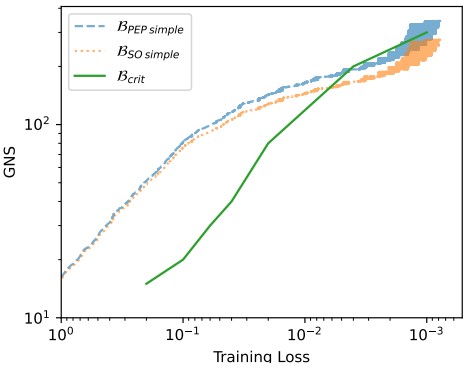

(a) Scatter plot comparing the exact and approximate GNS estimators $\mathcal{B}_{\text{PEPsimple}}$ and $\mathcal{B}_{\text{SOsimple}}$.

(b) Replication of GNS vs. loss plot from McCandlish et al. (2018), including their results for $\mathcal{B}_{\text{crit}}$ and both $\mathcal{B}_{\text{PEPsimple}}$ and $\mathcal{B}_{\text{SOsimple}}$.

Figure 2: Investigation of the accuracy of the approximation from Section 3.1 on MNIST.

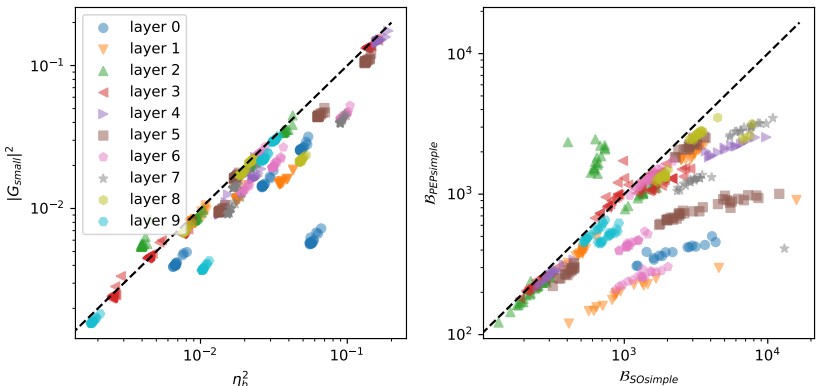

Figure 3: Results of a 111M parameter language model experiment measuring GNS on a fixed checkpoint. On the left, the approximate small batch gradient norm is compared to the exact and on right, the approximate SOGNS is compared to the exact PEPGNS.

## 4.2 Large Scale Gradient Noise Scale

To verify that this method is useful in practice, a checkpoint from a 111M parameter language model (Dey et al., 2023) was tested. In Figure 3, SOGNS and PEPGNS are compared, showing that the approximation tracks the exact case but diverges for some layers in the network. McCandlish et al. (2018) observes that the GNS may diverge by an order of magnitude from the measured *critical batch size* so the relationship we observe is within the margin of error.

## 5 Conclusion

Choosing a batch size is often achieved with reference to previous experiments or by hyperparameter search, which can be especially onerous in novel settings where a reasonable choice for batch size is not obvious. The GNS is a useful metric to navigate in such circumstances. In this paper, we observe that the per-example gradient norm trick (Goodfellow, 2015) could provide a useful shortcut for a minimal variance estimate of the GNS but it is inefficient in practical settings involving large transformer models (Li et al., 2022), requiring $O(T^2)$ operations in sequence length $T$. To address this, we propose SOGNS, an approximation that operates in $O(T)$, while correlating closely with the exact GNS. As practitioners now know that it is critical to log the gradient norms during training, we hope that this work can make GNS an accessible metric for large scale experiments.

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

# A  Taxonomy

The following taxonomy describes the different methods available to compute GNS. Each computes $|G_{B_{\text{small}}}|^2$ in a different way:

- Microbatch: multiple $G_{B_{\text{small}}}$ are computed over a set of microbatches
  - DDP: Each $G_{B_{\text{small}}}$ are gradients communicated between DDP nodes (McCandlish et al., 2018)
  - Sequential: Each $G_{B_{\text{small}}}$ are computed sequentially during gradient accumulation
- Subbatch: During gradient accumulation, select $G_{B_{\text{small}}}$ partway through
- Per-example:
  - Exact: $|G_{B_{\text{small}}}|^2$ is computed directly by the per-example gradient trick (Goodfellow, 2015; Li et al., 2022)
  - Approximation: $|G_{B_{\text{small}}}|^2$ is approximated by assuming input activations are normally distributed with mean zero; the focus of this work

The choice of which method to use may be dictated by the hardware available.

# B Variance of GNS Measurements

The GNS is a ratio estimator (Graunt, 1676), it is of the form $r = \frac{\bar{x}}{\bar{y}}$, where $\bar{x}$ and $\bar{y}$ are the sample means of two random variables, in this case $|\mathcal{G}|^2$ and $\mathcal{S}$.

To estimate the variance of this estimator we chose a Jackknife estimator (Choquet et al., 1999),

$$var(r) = \frac{n-1}{n} \sum_{i=1}^{n} (r_i - r_J)^2 \,,$$

where $r_i$ is the ratio estimator computed with the $i$th sample removed and $r_J$ is the jackknife estimate of the ratio. Performing a simulation with this estimator it is possible to estimate the effect of the $B_{\text{small}}$ and $B_{\text{big}}$ on the variance of the estimator. These two cases are illustrated in Figures 1. We can see that the size of $B_{\text{big}}$ is not important because the decrease in the variance as the number of samples increases is constant for all $B_{\text{big}}$. However, the size of $B_{\text{small}}$ is important because the variance decreases as $B_{\text{small}}$ increases, regardless of the samples processed.

This reinforces the intuition that the lowest variance estimate of the GNS should use the smallest $B_{\text{small}}$ possible. The smallest choice is $B_{\text{small}} = 1$, therefore obtaining the per-example gradient norm is valuable.

## B.1 Variance of the Gradient Noise Scale

The following code was used to produce Figure 1.

```python
import numpy as np
import matplotlib.pyplot as plt
import hashlib

from dataclasses import dataclass

N = 1000
scale = 1.
# use explicit random state, but set it to be random by default
rng = np.random.RandomState(np.random.randint(1))
true_G = rng.randn(N)
true_G = np.sqrt(N) * (true_G / np.linalg.norm(true_G)) # normalise to have exactly norm N

def draw_G(B):
    return (scale/np.sqrt(B)) * rng.randn(N) + true_G

def mean_of_microbatches(small_batch, large_batch):
    # this is the normal setting, where you have a large batch and you split it
    # into small batches, computing the norm of each and the norm of the whole
    assert large_batch % small_batch == 0
    r = large_batch // small_batch
    G = np.array([draw_G(small_batch) for _ in range(r)])
    return np.mean(np.linalg.norm(G, axis=1))**2, np.linalg.norm(G.mean(0))**2

funcs = {'mean_of_microbatches': mean_of_microbatches}

def jackknife(x, y):
    n = len(x)
    if n == 1:
        return x[0] / y[0], np.nan
    x, y = np.array(x), np.array(y)
    r = np.mean(x) / np.mean(y)
    x = x.reshape(-1, 1).repeat(n, axis=1) * ~np.eye(n, dtype=bool)
    y = y.reshape(-1, 1).repeat(n, axis=1) * ~np.eye(n, dtype=bool)
    r_i = np.mean(x, axis=0) / np.mean(y, axis=0) # vectorised jackknife
    r_j = n * r - (((n - 1) / n) * r_i.sum())
    # variance
    var_r = ((n - 1)/n) * np.sum((r_i - r_j)**2)
    return r_j, np.sqrt(var_r)

def run_replicates(large_batch, small_batch, replicates, func_type='simple_norms'):
    for _ in range(replicates):
        G_small, G_large = funcs[func_type](small_batch, large_batch)
        G_est = (large_batch * G_large - small_batch * G_small) / (large_batch - small_batch)
        S_est = (G_small - G_large) / (1./small_batch - 1./large_batch)
        yield S_est, G_est

@dataclass
class Experiment:
    samples_processed: list
    B_est: list
    sigmaB: list
    S_est: list
    G_est: list
```

```
    @staticmethod
    def mean(experiments):
      samples_processed = experiments[0].samples_processed
      B_est = np.mean([e.B_est for e in experiments], axis=0)
      sigmaB = np.mean([e.sigmaB for e in experiments], axis=0)
      S_est = np.mean([e.S_est for e in experiments], axis=0)
      G_est = np.mean([e.G_est for e in experiments], axis=0)
      return Experiment(samples_processed, B_est, sigmaB, S_est, G_est)

def gather_data(large_batch, small_batch):
  S_est, G_est = [], []
  samples_processed, B_est, sigmaB = [], [], []
  for i, (s, g) in enumerate(run_replicates(
                    large_batch, small_batch, 100, func_type='mean_of_microbatches'
                    )):
    S_est.append(s)
    G_est.append(g)
    b, sigma = jackknife(S_est, G_est)
    samples_processed.append((i+1) * large_batch)
    B_est.append(b)
    sigmaB.append(sigma)
  return Experiment(samples_processed, B_est, sigmaB, S_est, G_est)

def gather_cached_data(large_batch, small_batch):
  def generate_hash(large_batch, small_batch):
    batch_str = str(large_batch) + "_" + str(small_batch)
    hash_obj = hashlib.sha256(batch_str.encode())
    small_hash = hash_obj.hexdigest()[:8]
    return small_hash
  # repeatedly call gather_data and cache the results to file
  from pathlib import Path
  import pickle
  # check if cache_dir exists
  cache_dir = Path('gns_var_cache')
  cache_dir.mkdir(exist_ok=True)
  gns_var_fpath = cache_dir / f"gns_var_cache_{generate_hash(large_batch, small_batch)}.pkl"
  # load data if we have any
  if gns_var_fpath.exists():
    with open(gns_var_fpath, 'rb') as f:
      cached_experiments = pickle.load(f)
  else:
    cached_experiments = []
  # and then compute more anyway
  experiment = gather_data(large_batch, small_batch)
  # append this to the data we have
  cached_experiments.append(experiment)
  # save the data
  with open(gns_var_fpath, 'wb') as f:
    pickle.dump(cached_experiments, f)
  return Experiment.mean(cached_experiments)

def plot_gns_var(large_batches, small_batches):
  # this function can be run repeatedly to improve the estimate of the stderr
  prop_cycle = plt.rcParams['axes.prop_cycle']
  colors = prop_cycle.by_key()['color']
  fig, ax1a = plt.subplots(1, 1)
  fig.set_figheight(6)
  ax1b = ax1a.twinx()
  for i, (large_batch, small_batch) in enumerate(zip(large_batches, small_batches)):
    e = gather_cached_data(large_batch, small_batch)
    color = colors[i]
    ax1a.plot(e.samples_processed, e.B_est,
          label=f'l={large_batch}, s={small_batch}', alpha=0.5, color=color)
    ax1b.plot(e.samples_processed, e.sigmaB,
          label=f'l={large_batch}, s={small_batch}', alpha=0.5, color=color, linestyle='dashed')
  ax1a.hlines(1.0, 0, e.samples_processed[-1], linestyles='dashed', alpha=0.7)
  ax1a.set_ylim(0.9, 1.1)
  ax1a.set_xlabel('Samples processed')
  ax1a.set_ylabel('Solid: Estimated gradient noise scale')
  ax1b.set_ylabel('Dashed: Standard error of estimated gradient noise scale')
  ax1a.set_xscale('log')
  ax1a.legend()
  plt.show()

# example usage
plot_gns_var([4, 8, 16, 32, 64, 128], [1] * 6)
```

# C  Efficient Per-Example Gradient Norm Notation

This is a description of the trick proposed by Goodfellow (2015) using Einstein and Lagrange notation.

For the weights $\mathbf{W} \in \mathbb{R}^{I \times K}$ of a linear layer, with inputs $\mathbf{X} \in \mathbb{R}^{B \times I}$ and outputs $\mathbf{Y} \in \mathbb{R}^{B \times K}$, the gradient of the loss $l$ is

$$\frac{\delta l}{\delta \mathbf{W}} = \frac{\delta l}{\delta \mathbf{Y}} \frac{\delta \mathbf{Y}}{\delta \mathbf{W}} = \mathbf{X}^T \frac{\delta l}{\delta \mathbf{Y}}$$

which can be expressed in Einstein[4] and Lagrange notation for a batch (left) or per-example (right) as

$$w'_{ik} = x_{bi} y'_{bk} \quad w'_{bij} = x_{bi} y'_{bk}$$

with the squared norm in either case being

$$n^2 = (w')^2_{ik} = w'_{ik} w'_{ik} \quad n^2_b = (w')^2_{bik} = w'_{bik} w'_{bik}$$

and the per-example case factorizing as

$$n^2_b = (w')^2_{bik} = x_{bi} x_{bi} y'_{bk} y'_{bk}.$$

So, it is sufficient to computed the squared norm of $\mathbf{X}$ and $\mathbf{Y}'$ for each example to obtain exact per-example gradient norms of linear layer weights.

### C.1 Per-Example Gradient Norms in 3D

For three dimensions, $\mathbf{X} \in \mathbb{R}^{B \times T \times I}$ and $\mathbf{Y} \in \mathbb{R}^{B \times T \times K}$, the sums do not factorize because the per-example gradient must be reduced over the $t$ dimension:

$$w'_{ij} = x_{bti} y'_{btk} \quad w'_{bij} = x_{bti} y'_{btk}.$$

In this case the resulting per-example norm is (Li et al., 2022)

$$n^2_b = (w')^2_{bij} = \left( \sum_t x_{bti} y'_{btk} \right)^2 = x_{bti} y'_{btk} x_{bui} y'_{buk}.$$

The contraction order is vital to the efficiency of this computation as

$$n^2_b = \sum_{t,u} \left( \sum_i x_{bti} h_{btu} \right) \left( \sum_k y'_{btk} y'_{buk} \right)$$

has quadratic complexity over $1 \le u, i \le T$ where $T$ is typically sequence length in language modeling. In these cases, specifically when $2T^2 > IK$ (Li et al., 2022), computing the per-example gradients explicitly before reduction is preferred:

$$n^2_b = \sum_{i,k} \left( \sum_t x_{bti} y'_{btk} \right)^2.$$

This operation can also be performed as a grouped convolution (Rochette et al., 2019), but the overall contractions hit the same complexity limits. In our experiments involving convolutional layers we unfold using *im2col* and then apply the method above when computing exact or approximate gradient norms of convolutional layers.

### C.2 Index of symbols

As they appear:

$\mathcal{B}_{\textbf{simple}}$ The simple gradient noise scale introduced in McCandlish et al. (2018).

$\mathcal{B}_{\textbf{crit}}$ The critical batch size introduced in McCandlish et al. (2018).

$\Sigma$ The covariance matrix of the gradients.

$G$ The gradients of the loss $L$ with respect to the parameters of the model $\frac{\delta W}{\delta L}$. When provided without a subscript this refers to the "true" gradients minus the noise from SGD. When with subscript, such as $G_{B_{\text{big}}}$ and $G_{B_{\text{small}}}$, these are the gradients measured in SGD with the associated batch size, as described in McCandlish et al. (2018).

---

[4]Using this convention sums are omitted, the presence of indexes on the other side of the equation indicates where sums occur, in other words: $w'_{ik} = x_{bi} y'_{bk} \equiv w'_{ik} = \sum_b x_{bi} y'_{bk}$ (Einstein, 1916). This allows implementation via einsum functions in many popular frameworks (Rocktäschel, 2018).

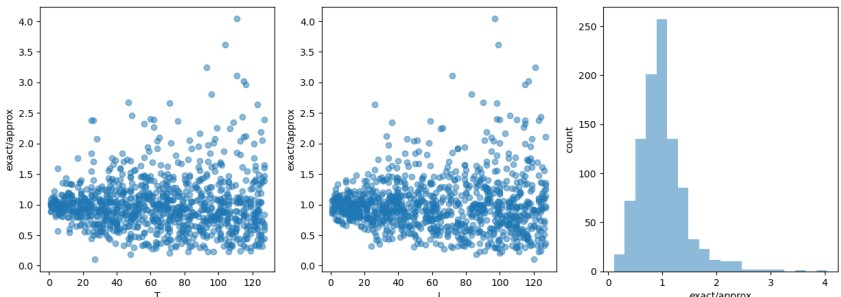

Figure 4: The naive approximation is compared to an exact computation of the per-example norm, with the ratio of the two shown on the y-axis.

$|\mathcal{G}|^2$ The unbiased estimator for the squared gradient norm introduced by McCandlish et al. (2018), defined in Equation 2.

$\mathcal{S}$ The unbiased estimator for the trace of the covariance matrix introduced by McCandlish et al. (2018), defined in Equation 2.

$B_{\text{big}}$ **and** $B_{\text{small}}$ The number of examples in the large and small batches, respectively.

$G_{B_{\text{big}}}$ **and** $G_{B_{\text{small}}}$ The gradients computed on the large and small batches, respectively.

$W$,$w$ The weights of a linear layer.

$\omega'$ Substitute weight gradient described in Appendix E.

$n_b$ Per-example norm of the gradient of the weights $w'$, indexed by the batch index $b$

$X$,$x$ Inputs to a linear layer.

$Y$,$y$ Outputs of a linear layer.

$\eta_b$ Approximation to the per-example norm $n_b$.

$\mathcal{B}_{\text{SOsimple}}$ The scaled output gradient noise scale as described in Section 3.1. This is computed using the approximate per-example norm $\eta_b$ as the small batch norm $|G_{B_{\text{small}}}|$ in the estimators of Equations 2.

**SOGNS** Acronym used to refer to $\mathcal{B}_{\text{SOsimple}}$ in the text.

$\mathcal{B}_{\text{PEPsimple}}$ The per-example parameter gradient noise scale, equivalent to $\mathcal{B}_{\text{simple}}$ when $B_{\text{small}} = 1$, uses the exact per-example gradient norm $n_b$ as $|G_{B_{\text{small}}}|$ in the estimators of Equations 2.

**PEPGNS** Acronym used to refer to $\mathcal{B}_{\text{PEPsimple}}$ in the text.

## D  Output Gradient Scaling Ablation

The approximation in Section 3.1 may either use the unit Gaussian assumption or assume the standard deviation of the activations is known; these are referred to here as the naive or relaxed assumptions, respectively. In the first case the output gradients are not scaled, in the second case they are scaled. The results of a simulation are shown in Figure 5 and Figure 4 for the relaxed and naive approximations. It can be seen that the relaxed approximation is more accurate than the naive approximation.

## E  Approximation of Large Batch Gradient Norms

The approximation presented in Section 3.1 may be interpreted as using a scaled version of the output gradient in place of the gradient with respect to the weights, specifically we can define $\omega'$ as

$$n_b{}^2 \approx \eta^2 = I\sigma_b{}^2 \sum_k \left( \sum_t y'_{btk} \right)^2 = \sum_k \omega'_{bk}{}^2 \quad \text{where} \quad \omega'_{bk} = \sqrt{I}\sigma_b \sum_t y'_{btk}.$$

The approximation can then also be applied to compute

$$|G_{B_{\text{big}}}|^2 \approx \eta^2 = \sum_k \left( \sum_b \omega'_{bk} \right)^2.$$

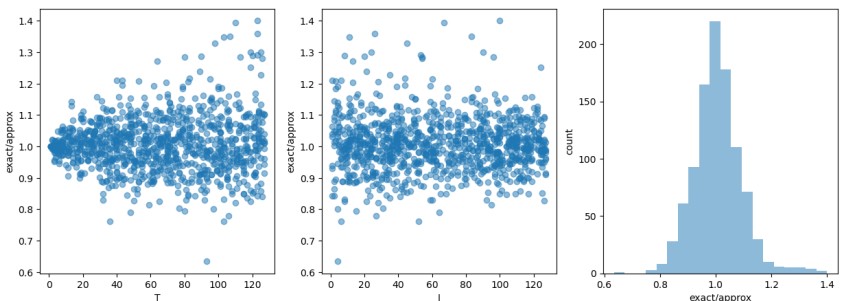

Figure 5: The relaxed approximation is compared to an exact computation of the per-example norm, with the ratio of the two shown on the y-axis.

The accuracy of this approximation is illustrated in Figure 6b.

# F MNIST Approximation Fit

For the remaining quantities not discussed in Section 4.1, Figure 6 describes the small batch squared gradient norm, the large batch squared gradient norm, the unbiased squared gradient norm and trace estimators of Equation 2. Code to replicate this experiment may be found (as linked in the main text), here: `https://gist.github.com/gaviag-cerebras/aa8050a2b4a2f327c83bc7b21f9e8b89`.

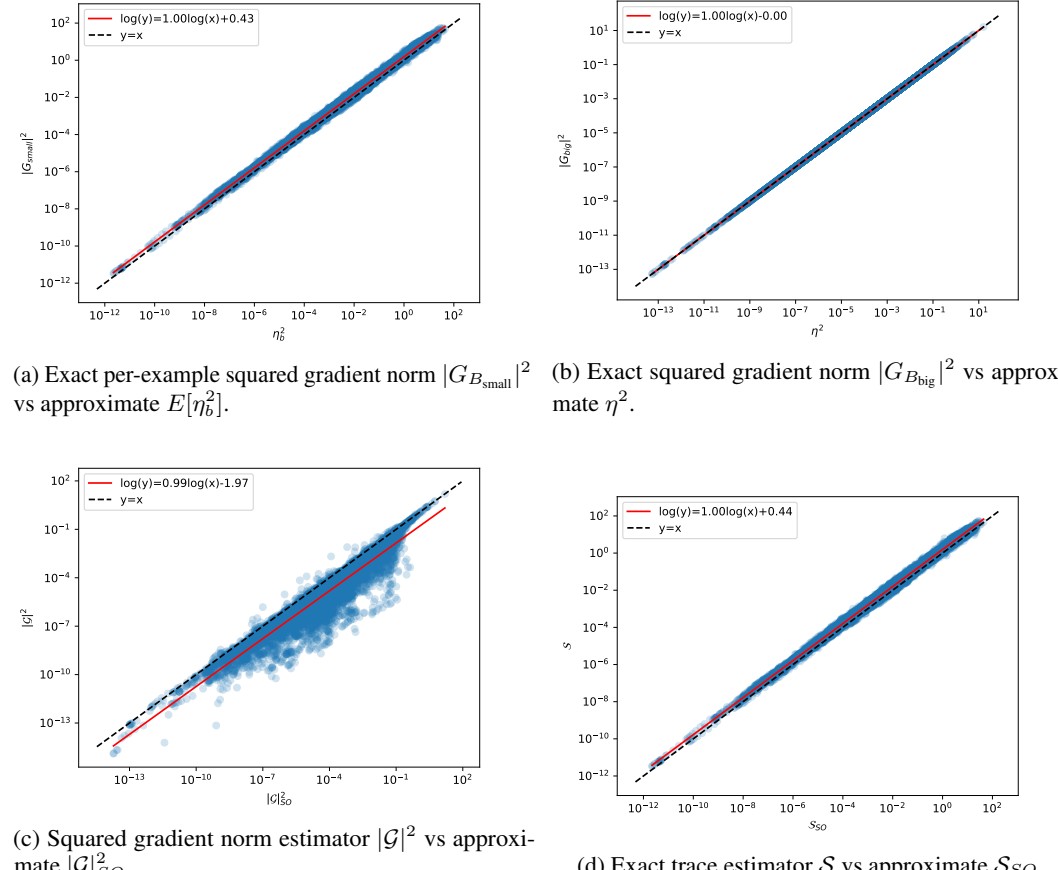

(a) Exact per-example squared gradient norm $|G_{B_{\text{small}}}|^2$ vs approximate $E[\eta_b^2]$.

(b) Exact squared gradient norm $|G_{B_{\text{big}}}|^2$ vs approximate $\eta^2$.

(c) Squared gradient norm estimator $|\mathcal{G}|^2$ vs approximate $|\mathcal{G}|^2_{SO}$.

(d) Exact trace estimator $\mathcal{S}$ vs approximate $\mathcal{S}_{SO}$.

Figure 6: Investigation of the accuracy of the approximation for all statistics discussed in Section 3.1 on MNIST, looking at only the convolutional layers.

