# OpenReview forum: "Efficient and Approximate Per-Example Gradient Norms for Gradient Noise Scale"
_NeurIPS.cc/2023/Workshop/WANT — WANT@NeurIPS 2023 Poster_

### Official Review · Reviewer_8UsB · 2023-10-23
**This paper suggests an interesting approximation method for gradient norms. Provided claims are not sufficient and paper misses key information for results reproduction. My recommendation: REJECT. .**

**Confidence:** 5

**Review:**

## Summary

This paper suggests a novel way to approximate the per-example gradient norm. Specifically, the authors build their assumption on the fact that layer inputs are distributed normally. They demonstrate a way in which gradient norms can be calculated for 3D inputs with quadratic complexity. The authors propose a method that is of linear complexity. The authors state that the practical value of the work is a simplification of gradient norm computation during training and by making it one of the metrics for evaluating large-scale experiments.

## Strengths

1. Writing is easy to follow, language is good
2. Mathematical grounding is coherent
3. Visual charts allow to better understand the concept and evaluate results


## Weaknesses

1. **Important information is missed**
   - no limitations section
   - no risks section
   - no ablation study

   Without this information, it is hard to evaluate the impact of the work on the current state-of-the-art. Which scenarios will not benefit from this new method? It is not a fundamental work, as it is stated in the paper to be devoted to practitioners.

2. **No way to reproduce results**
   - no link to code to reproduce reported results
   - no information on the number of examples, datasets, hardware setup
   - provided code is working for building 1 theoretical figure

   Thus, it allows to question reported results that might bring a wrong impression to future readers. I would highly recommend addressing this item in the future.

3. **Practical value of the work not clear**
   The only evidence stated in work is logging norms during training that *can be* used for monitoring training. No exact recommendation on how to use it as a metric. Nor there are examples of more clear benefits from the proposed approximations.

---

### Official Review · Reviewer_8Mvp · 2023-10-26
**this paper proposes an efficient approximation algorithm to compute gradient noise scale**

**Confidence:** 3

**Review:**

**Summary of the work**

This paper studies how to compute the gradient noise scale (GNS) more efficiently, which is important for determining an efficient batch size in deep learning models, especially in settings where large batch sizes are required for high-performance computing.
It proposes an approximation that assumes normally distributed activations at layer inputs, allowing for efficient access to per-example norms.

**Strength**

The paper proposes an efficient and approximate method, called Scaled Output Gradient Noise Scale (SOGNS), for estimating per-example gradient norms in deep learning models .

The SOGNS approximation seems to be generally applicable and provides accurate per-example gradient norms at negligible cost, making it suitable for practical settings.

The proposed method operates in O(T) complexity, making it more efficient than previous methods that require O(T^2) operations in sequence length T .

The paper provides empirical evidence and visualizations to support the accuracy and efficiency of the SOGNS approximation


**Weakness**

The paper is a bit hard to read, especially starting from the beginning of section 3. I think adding a table of main notations would mitigate this issue. For example, the authors never explain what $n_b^2$ is.

---

### Meta-Review · Area_Chair_rUsJ · 2023-10-27

**Recommendation:** Accept (Poster)
**Confidence:** 3

**Metareview:**

This manuscript proposes the idea of Scaled Output Gradient Noise Scale (SOGNS) to estimate per-example gradient norms in deep learning models.

The main concerns raised by the reviewers are the clarity and the potential practical value. Authors are encouraged to further improve the draft writing upon acceptance.

---

### Decision · Program_Chairs · 2023-10-28

**Decision:**

Accept (Poster)

**Comment:**

We thank the authors for their time and contribution to WANT and we are pleased to share that after the reviewing process the paper has been accepted. Congratulations! We encourage the authors to consider reviewers' feedback for the improvement of the camera-ready version. We hope to see you in person at the workshop and brainstorm on efficient training research together!